# Characteristics of a PVDF–Tin Dioxide Membrane Assisted by Electric Field Treatment

**DOI:** 10.3390/membranes12080772

**Published:** 2022-08-10

**Authors:** Muhammad Syahrul Nasution, Agung Mataram, Irsyadi Yani, Gurruh Dwi Septano

**Affiliations:** Mechanical Engineering, Engineering Faculty, University Sriwijaya, Palembang 30128, Indonesia

**Keywords:** polyvinylidene fluoride, tin dioxide, electric field, membranes, testing

## Abstract

Polymeric membranes have good properties for filtering water. In this paper, a membrane made from polyvinylidene fluoride (PVDF) polymer with 15 wt%, 17.5 wt%, and 20 wt% polymer content, with the addition of 1 wt% of tin dioxide with electric field treatment, is presented. The electric field used was DC 15,000 V. The membrane was tested to determine its characteristics and properties. The physical properties were examined with a scanning electron microscope, and the mechanical properties of the membrane were tested by tensile testing. The maximum tensile stress was obtained at 0.746 MPa, and the minimum tensile stress was obtained at 0.487 MPa. Microscopic examination of the membrane’s surface identified the shape, the structure of the fibers formed, and the amount of agglomeration. The flow rate, membrane flux, and normalized water permeability (NWP) were tested, using the water treatment performance test to measure the membrane’s filtering ability.

## 1. Introduction

Membrane technology has long been used as a method to filter water. Researchers have developed membrane technologies for water filtration known for their performance and low cost [1]. New membranes with good physical and mechanical properties as well as better water treatment performance are still being developed, using various mixtures of materials and combinations of optimal configurations in terms of the level of membrane strength and membrane pore size [2,3]. The main objectives of membrane formation include the long duration of the membrane’s life and the ability of the membrane to filter out impurities from water [4]. Therefore, it is necessary to have water filtration parameters in order to achieve detailed models and optimal water filtration phenomena on membranes, so water quality is obtained in accordance with water quality standards, with better inner membrane resistance and filtering of water impurities [5,6].

Polyvinylidene fluoride (PVDF) is widely used as a polymer membrane in the field of water treatment. PVDF is a homopolymer and copolymer that has high resistance to gas and water [7]. PVDF membranes also have the advantages of high physical and mechanical properties, temperature stability, good chemical resistance, and hydrophobicity (contact angle > 90°) [8]. However, high hydrophobicity leads to membrane fouling, which can lead to lower separation processes accompanied by increased energy consumption [9] and a decreased quality of the recycled water [10,11]. Research from Mertens et al. showed that a decrease in the water permeability performance of the membrane occurred in PVDF membranes without the use of additional additives. Hence, the PVDF membrane has a high hydrophobicity to water at a higher wt% polymer [12]. Therefore, in this research, we added tin dioxide with an electric field treatment before printing, which resulted in an increase in the value of the water permeability, because tin dioxide as a hydrophilic material was able to reduce the hydrophobicity of the PVDF membrane [13]. Tin dioxide is a very effective compound for water utilization applications. Tin dioxide is resistant to high temperatures in the range of 500–1000 °C, is nontoxic, inexpensive, has abundant sources, and has high chemical resistance to acidic and alkaline media [14]. Tin dioxide also has a hydroxyl group, which is able to bind and absorb hydrogen molecules that bind to water [15]. However, the addition of too much tin dioxide will cause the large porosity of the membrane surface, which causes a decrease in the mechanical strength of the membrane. Considering a total conversion of the precursor salt in metal oxide, the expected content was approximately 10 wt% for tin dioxide with respect to the polymer [4]. The solvent used in the manufacture of the membrane was N,N-dimethylformamide (DMF), which is a versatile organic solvent with excellent miscibility with water and most organic solvents [16]. The membranes were made of 15 wt%, 17.5 wt%, and 20 wt% (wt% is considered the polymer content) each, with a concentration of 1 wt% tin dioxide [13,17,18].

Treatment using an electric field in the membrane manufacturing process is a development that was introduced by several researchers on several membrane mixtures, and it is applied in this study. Electric field treatment is able to react the particles so that the distribution of particles is even [19]. In the research on membranes with a mixture of titanium dioxide and sulfonated poly (ether ether ketone), the particles in the casting solution under the AC electric field led to a uniformly distributed pattern of titanium dioxide in the sulfonated poly (ether ether ketone) matrix, which improved the mechanical properties of the hybrid membrane [20]. The other membranes, such as polymer polyethersulfone (PES) and graphite oxide, show that electric field-assisted blended membranes increased the hydrophilicity and negative charge density of the surface, which was caused by the migration of GO under the DC electric field [21]. Therefore, the manufacture of polyvinylidene difluoride (PVDF) membranes with tin dioxide as a membrane mixing material, using a DC 15,000 V electric field, was expected to reduce the porosity and increase the strength and resilience of the membrane [19].

## 2. Materials and Methods

### 2.1. Materials

PVDF polymers, N, N-dimethylformamide (DMF) solvents, and tin dioxide additive substances were obtained from Sigma-Aldrich (Palembang, Indonesia) with no further purification processes. We used concentrations of 15 wt%, 17.5 wt%, and 20 wt% (wt% is considered the polymer content) each, with a concentration of 1 wt% tin dioxide. PVDF polymers were used in granule form with a size of 5 mm. Tin dioxide was used in powder form with 5 μm. All materials used in this membrane fabrication study are shown in Figure 1.

### 2.2. Membrane Preparation

The membranes were prepared in three forms of solution with different concentrations of wt% (wt% is considered the polymer content) material mixture in each specimen, namely, 15 wt%, 17.5 wt%, and 20 wt%, with the addition of 1% tin dioxide. The weight percentage is calculated by the total mass of the polymer, at about 50 g. So, the mass of PVDF for each concentration was 7.5 g, 8.75 g, and 10 g. The 1 wt% of tin dioxide was considered from the polymer content of 0.075 g, 0.08752 g, and 0.1 g mass of tin dioxide, respectively, for the solvent using N,N-dimethylformamide, which had a mass of 42.425 g, 41.1625 g, and 39.9 g for each specimen.

The dissolution of the PVDF polymer using N,N-dimethylformamide with the addition of tin dioxide was performed, using a magnetic stirrer for 30 min at 25 °C at a stirring speed of 200 rpm. Then, the temperature was raised to 60 °C with a stirring speed of 500 rpm until the PVDF completely dissolved, for up to 4 h [22]. The PVDF solution was placed into a special airtight bottle to be stored and was allowed to stand for some time to determine whether there were still polymer fibers or solvents that were not homogeneous. Then, the solution that was homogeneous was poured evenly onto a mold made of rectangular copper plates, which was supplied with 15,000 V DC electricity for 2 min. After that, the solution was immersed in a coagulation bath filled with water until it released from the mold. The scheme for making the PVDF and tin dioxide membranes is shown in Figure 2.

### 2.3. Membrane Characterization

The development of membranes as a separation technology has encouraged researchers to examine the relationship between the morphology and performance of the membranes. Our main goal was to choose a suitable membrane fabrication, so that the membrane specifications met the environmental characteristics that the membrane will encounter. The morphology and surface structure of the membrane were viewed using a scanning electron microscope (SEM EVO MA 10, Oberkochen, Germany) at universal laboratory, Lampung University, Indonesia. the membrane was prepared as small pieces for viewing to observe the distribution, pores, and structure for each specimen. For the strength of the membrane, a tensile test was carried out, using a Zwick Roell material testing machine (Type BT2-FR020TH.A60 manufactured by Zwick Roell (Ulm, Germany) to analyze the resistance and the ability of the membrane to withstand tensile loads [23]. The results of the tensile test were the tensile stress and displacement, which were converted by the tensile test equipment into tensile and elongation stress data.

The water filter performance of the membranes is determined by normalized water permeability (NWP) through the cleanliness of a cassette after cleaning. This method involves measuring the passage of clean water through the membrane under standard pressure and temperature conditions [24]. Before the test, the membranes were wetted in ultrapure water for 15 min. After that, water was initially pumped to give a pressure of 1 bar to the membrane. From the 1 bar pressure, the performance of the membrane to filter water was measured with a 50 mL measuring cylinder for 1 h. Permeability measurements were conducted at ambient temperature and at constant cross flow. The membrane flux was calculated by Equation (1):(1)$$Jv=\frac{V}{A\cdot t\cdot P}$$
where *Jv* is the membrane flux (L/m^2^·h), *V* is the volume permeate (L), *A* is the effective area of the membrane (m^2^), *t* is the time for membrane filtration (h), and *P* is the pressure through the membrane (bar).

The normalized water permeability (NWP) is the liter per m^2^ per h per bar, LMH/bar or L/m^2^·h·bar based on Equation (2) [25].
(2)$$NWP=\frac{\left(Litres\;per\;hour\;volumetric\;permeate\;flow\;\right)\cdot \left(\mathrm{TCF}\right)}{\left(TMP\;bar\right)\cdot \left({\mathrm{membrane}\;\mathrm{area}\;m}^{2}\right)}$$
*TMP* = ((*Pfeed* + *Pretentate*)/2 − *Ppermeate*), the module average transmembrane pressure. TCF = (viscosity of water at measured temperature)/(viscosity of water @ 25 °C), the temperature correction factor. The schematic process of normalized water permeability is shown in Figure 3.

## 3. Results and Discussion

The results of the membrane test show the character and ability of the membrane to filter. Parameters, such as the mechanical properties, physical properties, and water treatment performance, identify the type of membrane separation process [26].

### 3.1. Membrane Structure

The functioning of the membrane depends on its structure, as this essentially determines the mechanism of separation and, thus, the application. The results of the appearance of the PVDF and tin dioxide membrane structure for each concentration are shown in Figure 4, Figure 5 and Figure 6.

The PVDF and tin dioxide membranes were observed from the top section, bottom section, and cross section. From the results of the microscopic testing using a scanning electron microscope (SEM), the shape of the membrane surface at each concentration was observed to be a sheet with micro-sized dimensions. Observing the top surface, the PVDF and tin dioxide membranes showed a smooth surface; the smooth surface of the membrane is able to reduce deposits that allow bacteria to gather, which reduces the biofouling during the water filtration process [27]. As for the bottom surface of the membrane, the surface structure comprised a multilayer structure. In the cross section of each concentration was an asymmetrical surface that formed the fiber braid; an increase in the wt% polymer caused the membrane structure of the fiber braid to be more tightly formed, creating smaller porosity in the membrane [28].

At each concentration of the membrane, there was a slight agglomeration around the membrane. At a concentration of 15 wt% agglomeration, this occurred more than in the other specimens; this affected the strength of the membrane because it became a stress concentrator, and the pore size that was not tight affected the reduction in the membrane area when tensile testing was carried out [29]. Agglomeration also reduces the interphase concentration, which causes weak modulus and strength. However, this was overcome by increasing the wt% polymer of the membrane, so that the viscosity increased and the sedimentation particles were also reduced [30]. The membrane particles were randomly distributed, which led to a small potential for agglomeration, because the treatment process using an electric field contributed to dispersing the material particles of the membrane [21].

### 3.2. Mechanical Properties

In testing the mechanical properties of the membrane, the standard specimens were used, ASTM (American Society of Testing Material) with Designation D638, with a specimen thickness of 3.2 ± 0.4 mm, an average speed of clamp 5 mm/minute, and at room temperature (25 °C) [31]. These methods determined the elastic, plastic, viscoelastic, hardness, fracture, and toughness properties [19,20]. The mechanical properties of the PVDF and tin dioxide membranes are presented in Table 1, and the tensile strength–elongation graph is shown in Figure 7, Figure 8 and Figure 9.

As shown in Figure 7, Figure 8 and Figure 9, the tensile strength of the membranes started at 0.10 MPa; this was because the membranes had characteristic elastoplastic material with linear strain hardening material. This behavior comprised two different parts. First, in the initial part of the curve, the behavior was rigid; therefore, the deformations were null until the yield stress $\sigma_{0}$ was reached. Above the yield stress, the generated strains were plastic (permanent) and proportional to the difference between the current stress and yield stress.

The mechanical model that describes this behavior is a parallel configuration connection between a linear spring with constant E and a friction plane with constant $\sigma_{0}$. With this configuration, the strain levels reached by both elements are the same, while the load (stress) applied to the system is distributed between both elements. If the stress is under reference $\sigma_{0}$, then the friction plane supports the entire load with null deformation. Once the yield stress is reached, the load on the friction plane remains at $\sigma_{0}$, while a deformation proportional to the load on the spring (σ −$\sigma_{0}$) is generated. When it is plastic, the strain generated remains permanent.

As shown in Figure 7, Figure 8 and Figure 9, the maximum tensile stress was obtained at a concentration of 20 wt% with a value of 0.746 MPa, and the minimum tensile stress for the PVDF and tin dioxide membranes was obtained at a concentration of 15 wt% with a value of 0.487 MPa. The maximum elongation was obtained at a concentration of 20 wt% with 3.74%, and the minimum elongation was obtained at a concentration of 15 wt% with 2.02%. There was an increase in the tensile strength of 36% and elongation value of 54%, with an increase in the wt% of PVDF with 1% tin dioxide.

Increasing the concentration affects the creep that occurs on the membrane [27], and membranes with higher concentrations tend to have less elongation; this is due to the influence of the increasingly dense fiber braid, which is able to withstand tensile stresses during the test, so that at break, the membrane breaks as a whole without any creep occurring [32].

Figure 7, Figure 8 and Figure 9 also show the viscoelastic behavior with various elongations [33]. Viscoelastic materials are capable of resisting shear flow (shear flow in solids and shear stress force gradients through the body) and strains, where the deformation of the material caused by stress is linear with stress. Shear stress is a stress state in which the stress is parallel to or tangent to the surface of the material, as opposed to the normal stress, which is perpendicular to the surface [34].

### 3.3. Membrane Permeability Performance

The membrane permeability performance can be directly examined through characterization was tested, using normalized water permeability (NWP). Due to the relatively heterogeneous nature of membrane permeability, membrane testing was carried out with several repetitions (n-run) [35]. The results of the membrane permeability performance test are presented in Table 2.

In the NWP test, the pressure used was 1 bar, which was classified into the type of microfiltration membrane [36]. The normalized water permeability of the membrane specimen of 15 wt% was 3.695 L/m^2^·h·bar; for the membrane specimen of 17.5 wt%, the NWP decreased to 1.749 L/m^2^·h·bar; and for the membrane specimen of 20 wt%, the NWP was 0.406 L/m^2^·h·bar. The graphs of the NWP are shown in Figure 10.

Figure 10 shows that the normalized water permeability membrane with PVDF 15 wt% with 1 wt% tin dioxide obtained the highest NWP with 3.695 L/m^2^·h·bar, and the membrane with the lowest NWP was the PVDF 20 wt% with 1 wt% tin dioxide with 0.406 L/m^2^·h·bar. There was a decrease in the ability of the membrane to carry out membrane filtration along with the increase in the wt% of the polymer. Figure 10 shows that the pores in the selective layer of the 15 wt% with 1 wt% tin dioxide membrane were quite open, as was already observed in Figure 4. This open separation layer led to very low retention of the dyes and high solvent permeability with this membrane. Slightly better separation results and a comparatively lower NWP were found for 20 wt% with 1 wt% tin dioxide [37]. This might be attributed to the formation of smaller membrane fiber sizes in high wt% polymer. As the wt% of polymer increased, the viscosity of the membrane solution increased, which inhibited the growth of the membrane pores. As a result, this contributed to a reduction in NWP [38]. In a study conducted by Silvia et al. using a carbon nanotube (CNT), an increase in wt% decreased the NWP, which then increased the tortuosity factor and blocked the passage of water molecules through the pores, thereby reducing the solvent permeation [39].

## 4. Conclusions

Based on the results and discussion of the development of a polyvinylidene fluoride (PVDF) membrane using tin dioxide with electric field treatment, the membrane was successfully developed with a 15,000 V DC electric field treatment. The SEM images showed an asymmetrical surface. The membranes with a higher wt% of polymer had a structure with a tighter fiber braid. The reaction of the tin dioxide particles with the PVDF polymer on the membrane also caused smaller porosity and agglomeration, which improved the mechanical properties of the membrane. The result of the tensile test showed that membranes with higher wt% of polymer were impacted by the tensile stress and elongation of the membrane. The addition of tin dioxide made the membrane sturdy with good stability to bind PVDF, which improved the viscoelastic properties of the membrane, enhancing the ability to absorb energy, flex, and spring back without cracking. Finally, with the water treatment performance, using normalized water permeability, there was a decrease in the ability of the membrane to carry out membrane filtration along with the increase in wt% of polymer. This might be attributed to the formation of smaller membrane fiber sizes in a higher wt% of polymer. As the wt% of the polymer increased, the viscosity of the membrane solution increased, which inhibited the growth of the membrane pores. As a result, this contributed to a reduction in NWP. The PVDF with the addition of 1 wt% tin dioxide with an electric field-assisted method is comparable to a wide range of filters for improved distribution in a polymer matrix.

## Figures and Tables

**Figure 1 membranes-12-00772-f001:**
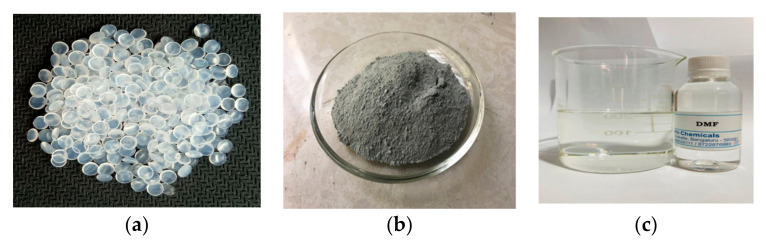
Material in membrane fabrication: (**a**) PVDF, (**b**) tin dioxide, and (**c**) DMF.

**Figure 2 membranes-12-00772-f002:**
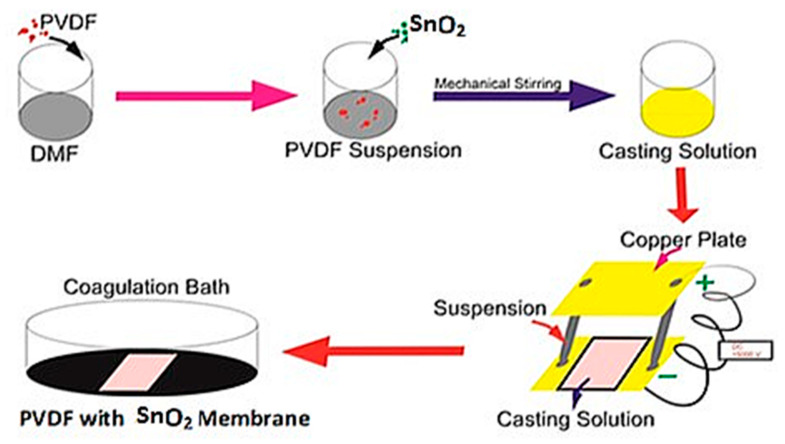
PVDF and tin dioxide membrane manufacturing process.

**Figure 3 membranes-12-00772-f003:**
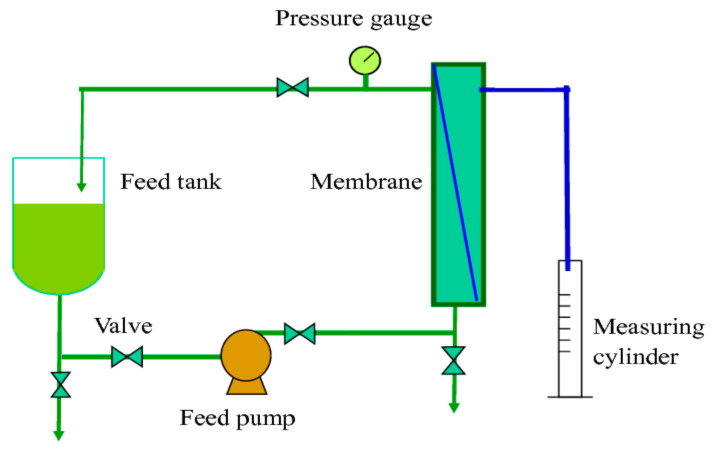
Schematic process of normalized water permeability.

**Figure 4 membranes-12-00772-f004:**
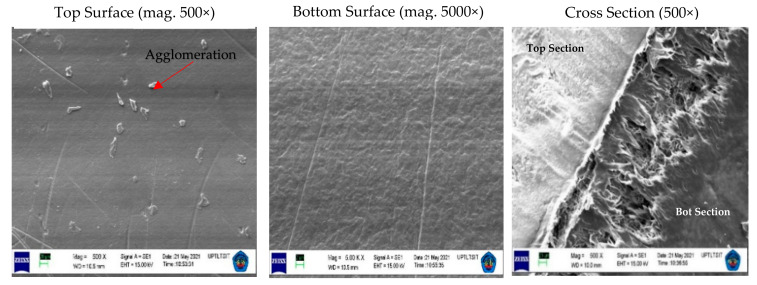
Structure of the PVDF and tin dioxide membrane for 15 wt% PVDF with 1% tin dioxide. Top Section with Magnitude = 500×, WD = 10.5 mm, EHT = 15.00 kV, Signal A = SE1. Bottom Section with Magnitude = 5.00 K ×, Working Distance (WD) = 10.5 mm, EHT = 15.00 kV, Signal A = SE1. Cross Section with Magnitude = 500×, WD = 10.0 mm, EHT = 15.00 kV, Signal = SE1.

**Figure 5 membranes-12-00772-f005:**
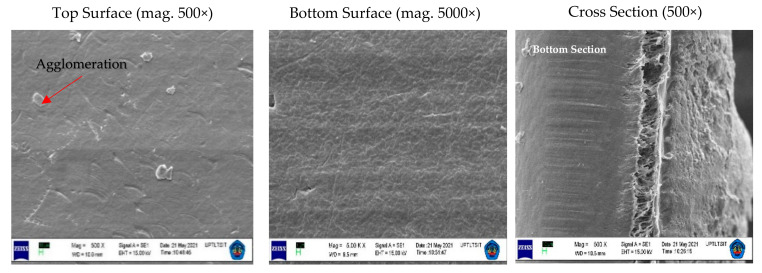
Structure of PVDF and tin dioxide membrane for 17.5 wt% PVDF with 1% tin dioxide. Top Section with Magnitude = 500×, WD = 10.0 mm, EHT = 15.00 kV, Signal A = SE1. Bottom Section with Magnitude = 5.00 K ×, Working Distance (WD) = 9.5 mm, EHT = 15.00 kV, Signal A = SE1. Cross Section with Magnitude = 500×, WD = 10.5 mm, EHT = 15.00 kV, Signal = SE1.

**Figure 6 membranes-12-00772-f006:**
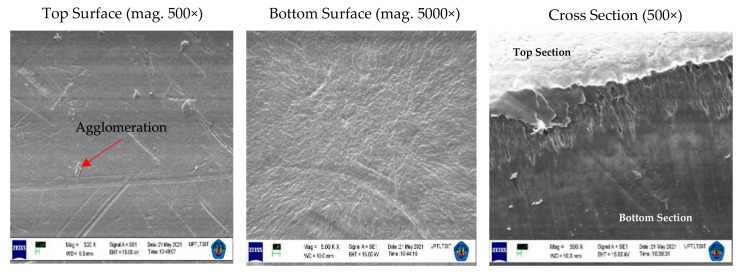
Structure of PVDF and tin dioxide membrane for 20 wt% PVDF with 1% tin dioxide. Top Section with Magnitude = 500×, Working Distance (WD) = 10.0 mm, EHT 15.00 kV Signal A = SE1, Bottom Section with Magnitude = 5.00 K ×, WD = 9.5 mm, EHT = 15.00 kV, Signal A = SE1. Cross Section with Magnitude 500×, WD = 10.5 mm, EHT = 15.00 kV, Signal SE1.

**Figure 7 membranes-12-00772-f007:**
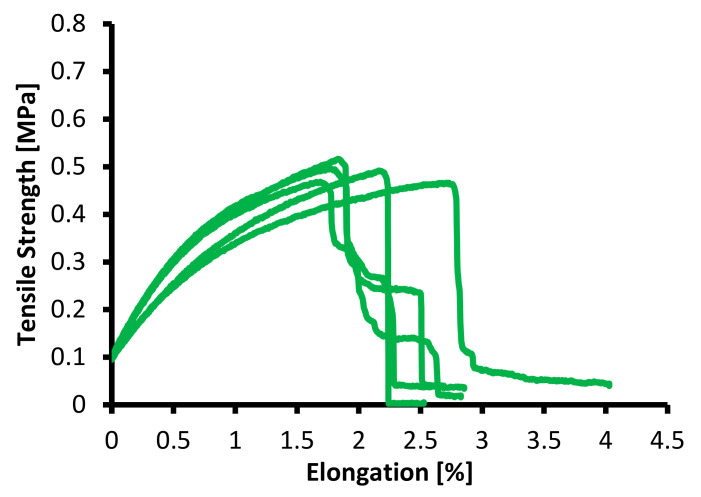
Tensile strength–elongation graph for 15 wt% PVDF with 1% tin dioxide.

**Figure 8 membranes-12-00772-f008:**
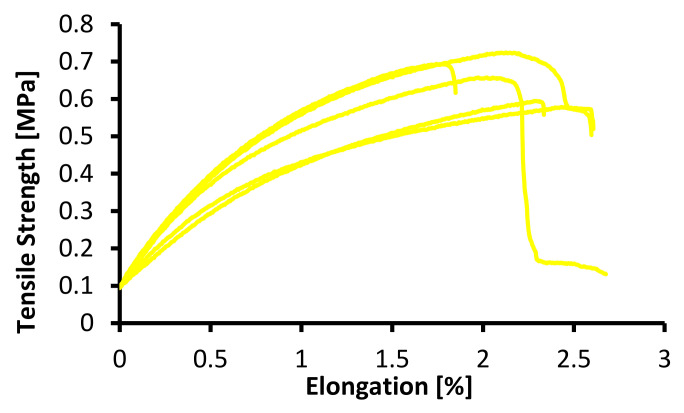
Tensile strength–elongation graph for 17.5 wt% PVDF with 1% tin dioxide.

**Figure 9 membranes-12-00772-f009:**
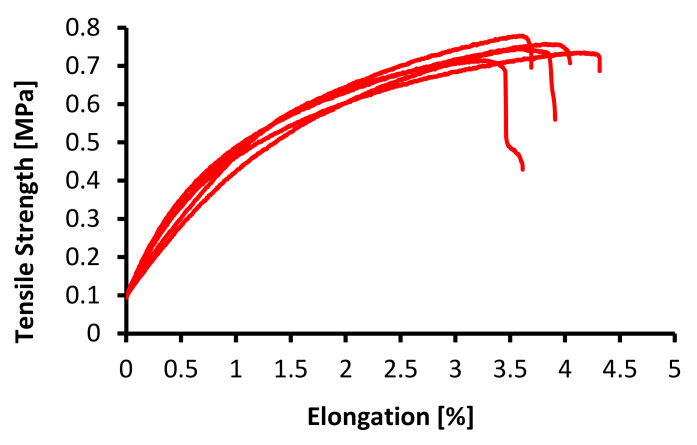
Tensile strength–elongation graph for 20 wt% PVDF with 1% tin dioxide.

**Figure 10 membranes-12-00772-f010:**
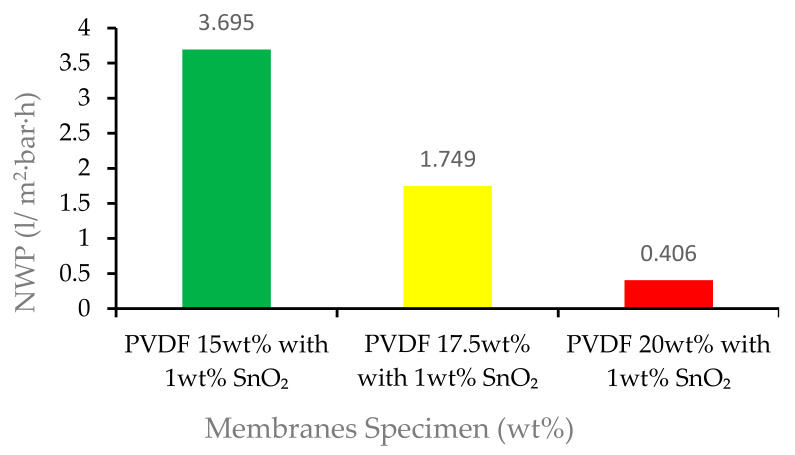
Average NWP of 15 wt%, 17.5 wt%, and 20 wt% PVDF with 1% tin dioxide membranes.

**Table 1 membranes-12-00772-t001:** Mechanical properties of the PVDF and tin dioxide membranes.

Membrane Specimen	Samples	Yield Strength (MPa)	Elongation at Break (%)
15 wt% PVDF with 1% tin dioxide	A1	0.468	1.68
	A2	0.496	1.76
	A3	0.492	2.16
	A4	0.466	2.70
	A5	0.516	1.83
Average		0.488	2.03
Median		0.494	1.99
Range		0.050	1.02
Standard Deviation		0.0209	0.4187
17.5 wt% PVDF with 1% tin dioxide	B1	0.657	1.97
	B2	0.693	1.74
	B3	0.724	2.12
	B4	0.578	2.43
	B5	0.594	2.28
Average		0.649	2.11
Median		0.657	2.12
Range		0.146	0.69
Standard Deviation		0.062	0.418
20 wt% PVDF with 1% tin dioxide	C1	0.714	3.57
	C2	0.745	3.57
	C3	0.779	3.61
	C4	0.757	3.82
	C5	0.735	4.17
Average		0.746	3.75
Median		0.745	3.61
Range		0.065	0.60
Standard Deviation		0.024	0.25

**Table 2 membranes-12-00772-t002:** Membrane permeability performance of the PVDF and tin dioxide membranes.

Membrane Specimen	Samples (*n*)	NWP (L/m^2^·h·bar)
15 wt% PVDF with 1% tin dioxide	1	3.837
	2	4.030
	3	3.942
	4	3.713
	5	2.953
Average		3.695
Median		3.837
Range		1.077
Standard Deviation		0.431
17.5 wt% PVDF with 1% tin dioxide	1	1.850
	2	1.719
	3	1.652
	4	1.690
	5	1.835
Average		1.7492
Median		1.7190
Range		0.1980
Standard Deviation		0.0886
20 wt% PVDF with 1% tin dioxide	1	0.402
	2	0.410
	3	0.405
	4	0.407
	5	0.405
Average		0.406
Median		0.405
Range		0.0079
Standard Deviation		0.0029

## Data Availability

Data are contained within the article.

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
