# Peer review of "Characteristics of a PVDF–Tin Dioxide Membrane Assisted by Electric Field Treatment"

_membranes, 2022, doi:10.3390/membranes12080772_

Round 1
Reviewer 1 Report
This paper mainly studied the characteristics of PVDF-Tin Dioxide membrane with 15,000 V DC electric field treatment. The overall framework of this paper is clear, and the characteristics of the membrane are analyzed mainly from three aspects. First, the physical properties of the prepared films were observed by SEM images, which indicates the reaction of tin dioxide particles with PVDF polymer on the membrane reduces the porosity and aggregates to improve the mechanical properties of the membrane. And then, the data from the tensile test show that the addition of tin dioxide enhances the ability of the film to absorb energy, bend and rebound without cracking. Finally, as the wt% of polymer increases, it can contribute to the reduction in NMP. I recommend publishing it in this journal after addressing the questions below.
1) Abstract, authors are suggested to start broad in the general background, then narrow in on the relevant topic that will be pursued in the paper. Maybe this part can be improved!
2) Introduction, authors are suggested to supplement relevant research status and provide a complete description of the main research content of the study in the last paragraph.
3) Introduction, in the second paragraph, "chemical resistance" is repeated.
4) Materials and Methods, in the third part, “TCF is the viscosity at measured temperature)/(viscosity of water at 25 °C)” lacks a parenthesis.
5) Results and Discussion, in the fourth paragraph of the second part, the values of tensile stress and elongation described do not correspond to those in the table 1.
Reviewer 2 Report
The problem taken up by the authors is very interesting.
However, the research presented, in my opinion, needs to be discussed in more details, and several issues need to be clarified and completed.
1. In formula 2, TMP should be precisely defined. As in the experiments presented in the paper, the following pressures were determined: pfeed pretentate and ppermeate ?
2. The tests should be completed by determining the contact angle for each membrane.
3. Why were the SEM images shown taken at different magnifications? What is the average size of the aggregates? Has their chemical composition been examined? The images should include a scale marker.
4. The results presented in Tables 1-2 and Figure 10 need to be supported by statistical analysis.
5. Not all abbreviations are explained in the text, e.g. WD (figures 4-6).
6. Bibliographic data of reference item No. 25 is insufficient. The title of the book and the publisher are missing.
Author Response
please see the attachment

This manuscript is a resubmission of an earlier submission. The following is a list of the peer review reports and author responses from that submission.
Round 1
Reviewer 1 Report
see attached file

Reviewer 2 Report
The authors report on formation of a membrane from Polyvinylidene Fluoride polymer with addition of Tin Dioxide. They observed the membrane with scanning electron microscope and measured its yield, limits of plasticity, tensile strength, elongation, and permeability. They presented data for membranes with different amount of polymer.
I suggest that the authors consider the comments below:
Page 2, Line 6: The membranes were made with three concentrations of 15wt%, 17.5wt% and 20wt% each with a concentration of 1wt% tin dioxide [15]–[17]. Please state the substance corresponding to 15wt%, 17.5wt% and 20wt%.
The description of the membrane preparation should be clear enough to allow the readers to repeat the procedure. Besides about 20% of the polymer, what substance was used to produce the membrane? How much material was used? What laboratory material was used? Please state clearly by using simple and short sentences.
The sentence “Dissolution of PVDF polymer using N,N-Dimethylformamide with addition of Tin Dioxide (SnO2) using a magnetic stirrer.” Is missing the verb.
Next sentence: …the solution that is considered homogeneous poured evenly on…in missing the verb.
Next sentence: “After that it is immersed …” Please replace “it” with the subject.
Page 3, Line 10: “… for each membrane concentration” I do not understand what is “membrane concentration” Please rephrase or explain or omit.
Page 3, Paragraph 2, Line 6: “until reach 50 ml.” Please correct the syntax.
Eq. 1: the symbol for pressure should be capital (P) to match the one in the legend. Also, the italic/roman font in the equation and in the text should be matched.
Membrane characterization
Hpw were the samples prepared for SEM?
The authors should briefly describe the essentials of the methods used by the equipment that they applied.
Membrane structure: some basic data on the membrane would be welcome: what is the thickenss of the membrane, what area sizes are usually utilized. Authors could present a photo of the membrane. Figures 3-5. A unit length bar with the information of the respective size would be of help to the reader. The text in the caption “Membrane Structure PVDF and Tin dioxide for 15wt% PVDF with 1% SnO2” is unclear. Please correct the syntax. The composition of figures should be improved. Composition of panels in Figure 3 is slopy. Panels should be divided by lines. Markings and descriptions (such as “Top surface”) should be made parts of the figures or of the captions. Panels should be marked (for example A,B,C, etc.) and referred to in the text.
The text below Figure 5:
“The shape of the membrane surface at each concentration is shaped like a sheet with nanometer size dimensions”. Please rephrase. How this can be seen from the image with magnification 500X? If I am correct, the whole image would be around 100 micrometers long.
“The surface structure is shaped like a radius” is unclear to me. Please explain and refer to the figures. If necessary, please equip images with some indicators such as arrows.
“The increase in the concentration of the PVDF and Tin dioxide membrane is getting higher, making the structure of the fiber braid that is formed also tighter”. Should there be evidence in the images? Please give arguments or give quantitative data. The cross sections are not presented from the same angle, so comparison is not straightforward. Indicate on the cross section images where is the top and where is the bottom. Indicate with an arrow where is the fiber braid.
Are there any differences between Figures 3, 4 and 5? Please indicate the differences or state that there are no differences.
“At each concentration of the membrane there is a slight agglomeration around the membrane.” What is the “concentration of the membrane”? Please indicate (e.g. with arrows) where are the agglomerates. Please correct the syntax. What is agglomerated?
The authors introduce terms “membrane particles”. Please explain what are these particles.
“The treatment process using an electric field contributes to dispersing the nanoparticles on the membrane”. The nanoparticles are here mentioned for the first time. Please explain. If this is a part of the results, please amend the Introduction with the respective information.
Mechanical properties: please explain the abbreviation ASTM D638. If this is an equipment please amend the Methods.
Table 1. Please give also the average values for each set of the data.
Figures 6-8. Comparison would be easier if the curves in Figures 6-8 were presented in the same graph (e.g. with different colors for different set of data). If the authors decide to keep separate graphs, it would be of help to the reader that the graphs have the same scales on both axes.
“The viscoelasticity of the membrane has the shortest elongation compared to other concentrations, with an elongation difference of 0.1%.” This seem unclear. Please clarify. Was the viscoelasticity measured? If not, please stay away from referring to it in Results.
“The effect of increasing concentration affects the creep that occurs on the membrane”. Please explain what is the substance referred to by “concentration”.
“Typically, polymers exhibit combined elastic and plastic deformation behavior de-pending on temperature and strain rate. At low temperature and high strain rate, elastic behavior is observed, and at high temperature but low strain rate, viscous behavior is observed. The combined behavior of viscosity and elasticity was observed at the interme-diate temperature and strain rate values.” Are these the results of the authors? If so, please give the methods and the data. If not, please give respective citations.
Membrane permeability Performance
“Membrane permeability performance was tested using Normalized Water Permeability (NWP).” Please describe the measurement in the Methods.
Table 2. Please give also the average values for each set of the data in the table. There is no need to repeat the numbers in the text.
“The flow rate that can be filtered by the membrane” is unclear to me. Please rephrase.
Figure 9: Please correct the syntax in “Flux Membranes of PVDF and Tin Dioxide” There is no need to give numbers in the graph indicating the values of the ordinate.
Figure 10: “Concentration of Membrane (wt%)” is unclear to me. Please rephrase.
The discussion lacks comparison of the authors’ findings with the findings of other authors. Please amend.
Conclusions seem more as a Discussion. I suggest that some parts of it are moved to the section “Results and Discussion”, in particular those with merely stating the numbers. Instead, the authors should outline these results and convince the readers about what is new in them with respect to previously gathered knowledge.
The English language should be improved. I suggest that the authors seek help from a native speaking scientist in order to formulate what they would like to convey to the readers of their work.
Reviewer 3 Report
Charateristic Of PVDF-Tin Dioxide membrane assisted by Electric Field Treatment is very interesting paper.Some improvement is required.
Page 1: Abstract: In this paper, the membrane was made by Polyvinylidene Fluoride (PVDF) polymer with addition of Tin Dioxide (particle size, ratio,..)
AbstracT: The physical properties (which properties) were examined using a scanning electron microscope,
0.746 Mpa (correct: 0.746 MPa)
Page 1: 500 ÌŠC - 1000 ÌŠC C, to delete C
Page 2: But the addition of too much (please to add a value for limitation) Tin Dioxide will cause a large porosity on the membrane surface
Page 3: P = Pressure (bar) or MPa
General questions:
- Can you give an information about particle size of used tin oxide
- In your text we can find sometimes an unit for pressure (MPa) and sometimes (bar). Can you use a same unit
- What is an influence of particle size of Tin oxide and a percentage of added tin oxide on physical and mechanical properties of membrane?
Round 2
Reviewer 1 Report
see attached

Round 3
Reviewer 1 Report
In the re-resubmitted manuscript Eq, 2 has been corrected. The mechanical response has been given the third type of description, still not acceptable. In addition, the overall content offered remains a very marginal contribution to the field.
In conclusion:
The paper does not contribute significant information to the field, it is of little interest and in addition still contains errors. I recommend rejection.